# High-Quality Genome Assembly and Genome-Wide Association Study of Male Sterility Provide Resources for Flax Improvement

**DOI:** 10.3390/plants12152773

**Published:** 2023-07-26

**Authors:** Xiaoqing Zhao, Liuxi Yi, Yongchun Zuo, Fengyun Gao, Yuchen Cheng, Hui Zhang, Yu Zhou, Xiaoyun Jia, Shaofeng Su, Dejian Zhang, Xiangqian Zhang, Yongfeng Ren, Yanxin Mu, Xiaolei Jin, Qiang Li, Siqin Bateer, Zhanyuan Lu

**Affiliations:** 1Inner Mongolia Academy of Agricultural & Animal Husbandry Sciences, Hohhot 010031, China; 2Agricultural College, Inner Mongolia Agricultural University, Hohhot 010019, China; 3College of Life Sciences, Inner Mongolia University, Hohhot 010070, China; 4State Key Laboratory of Reproductive Regulation and Breeding of Grassland Livestock, Hohhot 010019, China; 5Inner Mongolia Key Laboratory of Degradation Farmland Ecological Restoration and Pollution Control, Hohhot 010031, China; 6Inner Mongolia Conservation Tillage Engineering Technology Research Center, Hohhot 010031, China

**Keywords:** flax, genome assembly, male sterility, GWAS

## Abstract

Flax is an economic crop with a long history. It is grown worldwide and is mainly used for edible oil, industry, and textiles. Here, we reported a high-quality genome assembly for “Neiya No. 9”, a popular variety widely grown in China. Combining PacBio long reads, Hi-C sequencing, and a genetic map reported previously, a genome assembly of 473.55 Mb was constructed, which covers ~94.7% of the flax genome. These sequences were anchored onto 15 chromosomes. The N50 lengths of the contig and scaffold were 0.91 Mb and 31.72 Mb, respectively. A total of 32,786 protein-coding genes were annotated, and 95.9% of complete BUSCOs were found. Through morphological and cytological observation, the male sterility of flax was considered dominant nuclear sterility. Through GWAS analysis, the gene *LUSG00017705* (cysteine synthase gene) was found to be closest to the most significant SNP, and the expression level of this gene was significantly lower in male sterile plants than in fertile plants. Among the significant SNPs identified in the GWAS analysis, only two were located in the coding region, and these two SNPs caused changes in the protein encoded by *LUSG00017565* (cysteine protease gene). It was speculated that these two genes may be related to male sterility in flax. This is the first time the molecular mechanism of male sterility in flax has been reported. The high-quality genome assembly and the male sterility genes revealed, provided a solid foundation for flax breeding.

## 1. Introduction

Flax (*Linum usitatissimum* L.) is an annual herb belonging to the family Linaceae and the genus *Linum*. As an ancient crop, flax has been used in the Near East for more than 10,000 years [1]. It is cultivated as an economically important oil and fiber crop in cooler regions of the world [2,3]. The originally cultivated flax is supposed to be descended from a wild flax species, pale flax [4,5] (*Linum bienne* Mill.). Cultivated flax is well utilized for various purposes, such as edible oil, textile fiber, animal feed, and other industrial products [6].

Flax is a self-pollinating crop that requires manual emasculation in hybrid production, which is prone to flower damage and a decrease in yield. Therefore, it is difficult to cultivate special varieties with good comprehensive traits through hybridization. Hybrid breeding technology has been widely applied in crops such as rice and corn, resulting in a significant increase in yield [7]. The heterosis phenomenon also exists in flax [8,9]. The foundation of hybrid breeding lies in the search for male sterility lines [10]. For this reason, breeders have been looking for ways to take the benefit of heterosis in flax. The research on flax male sterility has been going on for many years. Bateson and Gairdner (1921) identified the flax cytoplasmic male sterile line for the first time [11]. However, the petals of this sterile line could not fully unfold, cross-pollination was hindered, and it was difficult to use for hybrid production. After that, Kumar [12] and Thomsom [13] reported flax cytoplasmic male sterile with fully expanded petals, but they were also not successful in its utilization for hybrid production. Dang et al. carried out research on antibiotic-induced flax male sterile lines and obtained new male sterile flax mutants [14], but there was no report on the effective utilization of heterosis. In 1986, Chen et al. discovered the naturally mutated dominant nuclear male sterile flax, which has the characteristics of fully open petals, obvious sterility traits, and stable sterility [15]. Our research group bred a dual-purpose line for both oil and fiber production with a stable sterility rate using this sterile material and created a hybrid with strong heterosis, which has high utilization value. The Neiya NO.9 cv. used in this study was also developed using the same male sterile line as the female parent.

High-quality reference genomes are the foundation of crop genetics research. The estimated genome size of flax is 373 Mb, which is diploid and consists of 15 chromosomes (2n = 30) [16,17]. Currently, several genomes of flax have been reported [16,17,18,19,20]. However, they were all assembled from short reads and consisted of thousands of contigs, which are very common in plants due to the high repeatability of the plant genome. Meanwhile, fragmented genome assemblies often contain partial genes, collapsed or redundant repeat sequences, and chimeric contigs that confound functional gene mapping. There is a high contiguous genome assembly that has been reported recently for fiber flax [19]. However, assembly using long-read sequencing technology is still lacking for oil flax. Thus, using PacBio long reads and Hi-C sequencing data, combined with a genetic map reported previously [21], we first reported the high-quality assembly of the oil flax genome. Using this high-quality genome, we also mapped the gene putatively responsible for nuclear male sterility in flax through a whole-genome association study.

## 2. Results

### 2.1. Genome Sequencing, Assembly, and Quality Evaluation

To estimate the genome size of flax Neiya No. 9 cv. (Figure 1a), 49.82 Gb of Illumina paired-end reads were utilized to generate 111 mer frequency data for the genome survey. This resulted in an estimated genome size of 506.4 Mb and a heterozygosity of 0.02% (Appendix A, Appendix A). The genome size estimate obtained through flow cytometry was similar (~500 Mb). PacBio long reads (55.72 Gb of subreads) were used to assemble the genome, resulting in 6,099 contigs and an N50 contig length of 0.91 Mb (total size of 473.55 Mb, Appendix A).

Hi-C paired-end reads (49.01 Gb of raw data) were then aligned to the assembled contigs using BWA-mem [22]. The 3D-DNA [23] pipeline was utilized to generate the Hi-C contact map. In conjunction with a previously reported genetic map [21], contigs were successfully anchored onto 15 chromosomes, effectively covering 97.6% of the overall length (Figure 1b). Among the chromosomes, the average length was 30.87 Mb, with Chr14 being the shortest at 19.91 Mb and Chr10 being the longest at 42.83 Mb (Appendix A). The entire genome spanned 474.08 Mb and comprised 836 scaffolds, with a scaffold N50 value of 31.72 Mb (Appendix A).

In order to assess the quality of the Neiya No. 9 genome assembly, we conducted mapping analysis of short paired-end reads obtained from whole-genome sequencing. The results revealed an overall alignment rate of 99.70%, covering 99.74% of the assembled sequences (Appendix A). This suggests that our assembly encompasses nearly all the genomic information present in the Neiya No. 9 genome. Additionally, we employed the BUSCO [24] gene set for eudicots (eudicotyledons_odb10), comprising 2121 single-copy orthologous genes, to evaluate the assembly. Our findings indicate that the Neiya No. 9 genome assembly contains 95.9% of the conserved genes (Figure 2a, Appendix A), slightly surpassing the CDC Bethune v2 assembly (93.5%) and the YY5 v2.0 assembly (94.4%) (Appendix A). Assembling a plant genome poses challenges due to the prevalence of repetitive sequences. Recently, a reference-free metric called the LTR Assembly Index (LAI) was introduced to evaluate assembly continuity utilizing LTR retrotransposons (LTR-RTs) [25]. To further evaluate the quality of our newly assembled flax genome, we computed the LAI score, resulting in a value of 15.83 (Figure 2c, Appendix A). This score significantly surpassed that of the CDC Bethune v2 assembly (LAI = 9.39) and the YY5 v2.0 assembly (14.29) (Appendix A), signifying a highly reliable reference genome assembly [25]. Furthermore, our observations indicate that intact LTR-RT insertion events have occurred more recently in the Neiya No. 9 genome compared to the CDC Bethune genome (Figure 2b). Collectively, these lines of evidence substantiate the exceptional quality of the assembled flax genome.

### 2.2. Genome Annotation

The Neiya No. 9 genome was found to contain a total of 187.16 Mb repetitive sequences, accounting for 39.5% of the genome (Figure 1b). Among these repetitive elements, transposable elements (TEs) constituted 28.9% of the repeats. Specifically, long terminal repeats (LTRs) were identified as the most abundant type of TE, occupying 21.4% of the genome (Appendix A). Additionally, our analysis predicted the presence of 168 miRNA genes, 1952 transfer RNA genes, 518 small nuclear RNA genes, 224 spliceosomal RNA genes, and 6978 ribosomal RNA genes within the assembly (Appendix A).

To predict gene structures, we employed an integrative strategy that combined protein-based homology searches and transcript data obtained from RNA-Seq analysis of flax roots, leaves, stems, flowers, and fruits. In total, 32,786 protein-coding genes were predicted for the flax genome (Figure 1b). Among these protein-coding genes, 29,546 (90.1%), 18,708 (57.1%), 18,107 (55.2%), 32,169 (98.1%), and 27,777 (84.7%) genes were successfully annotated using the Pfam, Gene Ontology (GO), Kyoto Encyclopedia of Genes and Genomes (KEGG), Non-Redundant (NR), and SwissProt databases, respectively (Appendix A). Overall, 32,613 (99.5%) genes were successfully annotated with at least one database (Appendix A).

The average lengths of transcripts and coding sequences (CDS) were calculated as 2458 bp and 210 bp, respectively. Moreover, the average lengths of exons and introns were determined to be 228 bp and 608 bp, respectively, with an average of 5.7 exons per gene. By mapping RNA reads onto the annotated genome, we found that the majority of the RNA reads (75.0%) from the five analyzed flax tissues could be aligned to annotated exon regions, corresponding to 32,183 genes.

### 2.3. Whole-Genome-Duplication Analysis

In plant evolution, whole-genome-duplication (WGD) events are widespread and have a significant impact on species diversification. In our study, we identified a total of 10,150 groups of homologous gene families across the flax genome. To further investigate the timing of these duplication events, we calculated synonymous substitution rates (Ks) for all pairs of homologous genes. The frequency distribution of Ks exhibited a distinct peak at Ks = 0.1174 and a secondary peak at Ks = 0.6497 (refer to Appendix A). These peaks suggest the occurrence of two genome-duplication events. The first, more recent duplication event is estimated to have taken place between 3.91 and 7.25 million years ago (MYA). The second, more ancient duplication event is estimated to have occurred between 21.66 and 40.11 MYA. These estimates are based on the assumption of a substitution rate ranging between 1.5 × 10^−8^ and 8.1 × 10^−9^ substitutions per synonymous site per year [26,27]. Our findings align with those of a previously published flax genome, with the notable difference that the assembly size of the Neiya No. 9 flax genome is approximately 127 MB larger than that of the CDC Bethune genome [16,17].

### 2.4. Comparative Genomics

In order to explore the evolution of the flax genome, we conducted comparative analysis with the genomes of eight other species: *Arabidopsis thaliana*, *Glycine max*, *Medicago truncatula*, *Populus trichocarpa*, *Salix brachista*, *Ricinus communis*, *Jatropha curcas*, and *Manihot esculenta* (Appendix A). By comparing flax with these species, we identified 2877 genes (belonging to 1155 families) that were unique to flax. To gain insights into the functional characteristics of these unique genes, we performed Gene Ontology (GO) enrichment analysis. The results revealed significant enrichment in various biological processes and molecular functions. Specifically, the unique genes were found to be enriched in lactase activity, cellobiose glucosidase activity, the very long-chain fatty-acid biosynthetic process, fatty-acid omega-oxidation, glucan exo-1,3-beta-glucosidase activity, diterpenoid metabolic process, and response to metal ion. These enrichments shed light on the specific functional adaptations and distinctive features of the flax genome compared to the other analyzed species.

Through further phylogenetic analysis, we were able to estimate the divergence time between flax and other plant species, indicating that flax diverged from other species within the order Malpighiales approximately 92 million years ago (as shown in Figure 2d). To understand the dynamics of gene family evolution, we examined the expansion and contraction of gene families by modeling gene gain and loss across the phylogenetic tree. Our analysis identified 5920 expanded genes and 8483 contracted genes in flax (as depicted in Figure 2d). Additionally, we identified a total of 30 gene families that were undergoing rapid evolution. The Gene Ontology (GO) classification of these rapidly evolving gene families revealed enrichment of genes involved in steroid hydroxylase activity, lipid homeostasis, and the brassinosteroid metabolic process.

### 2.5. Phenotypic Characteristics of Male Sterility in Flax

Sterile plants are divided into two types according to the shape of the stamens. One is that the petals are coiled around each other and cannot be unfolded, covering the stigma, and the petals are not easy to fall off; the other is normal flowering, with the stigma exposed (Appendix A). Compared with fertile plants, they all have thin anthers, no pollen, normal pistil development, and can be pollinated by foreign pollen. The flowers of sterile plants are light blue, and the seed coat is nearly white, while the flowers of fertile plants are dark blue, and the seed coat is brown. Every year before flowering, infertile plants were subjected to backcrossing, testcrossing, and open pollination. The phenotype of the resulting infertile flax offspring was continuously observed and statistically analyzed. It was found that the segregation ratio of fertility in the offspring all conformed to a 1:1 ratio (chi-squared test, *p* > 0.05) (Appendix A). It is speculated that the naturally mutated flax sterile material we found is dominant nuclear male sterility, and the sterile gene is lethal in the homozygous state.

The anthers of sterile plants at different developmental stages were observed by the paraffin section. It was found that the anther tapetum was separated from the middle layer, and the development of microspores stopped in the process of the completion of the first division of meiosis and the formation of tetrads in the second mitotic division of pollen mother cells of sterile plants, resulting in shrunken anther chambers. Fertile pollen mother cells undergo meiosis to form tetrads and then develop into mononuclear pollen grains and mature pollen. The tapetum gradually disintegrates in situ throughout development and disappears completely by the time of mature pollen grain (Figure 3).

### 2.6. Cysteine Protease May Play a Key Role in Male Sterility of Flax

To study the genetic mechanism of male sterility in flax, 20 sterile plants and 36 fertile plants were collected for whole-genome resequencing. The average sequencing depth was 9.4×, and the average coverage was 99.53%. A total of 2,255,667 high-quality SNPs were detected, and genome-wide association analysis was performed. A *p*-value of 2.2 × 10^−8^ was used as the genome-wide significance threshold. A total of 36 SNPs were significantly associated with the sterility-related trait; most of them were concentrated on chromosome 10 (Figure 4a). Among these SNPs, only 2 SNPs were located in coding regions and annotated as nonsynonymous. There were total of 32 loci included in a region of 18.8 Mb on chromosome 10 that may be associated with the sterility-related trait. Thirty-eight candidate genes were extracted from this region. Subsequently, GO enrichment analysis was employed to identify the major biochemical and signal transduction pathways in which the 38 candidate genes were involved. GO analysis revealed several biological processes enriched, such as cysteine synthase activity, the development of flowers, cell wall biogenesis, regulation of post-transcriptional gene silencing, and anther development, which are highly associated with the development of pollen. Among these candidate genes, *LUSG00017705*, the gene closest to the most significant SNP, is annotated as cysteine synthase, which is specifically expressed in the tapetum and causes dominant male sterility in many plants, such as rice and rapeseed [28,29].

In order to verify the candidate genes related to male sterility obtained by GWAS, we collected flower buds of sterile (MS2, MS3, MS5) and fertile (S13, S16, S21) plants at different stages (microspore mother cell stage, tetrad stage) of the same plant for RNA-Seq and detected a total of 3087 and 4683 differentially expressed genes, respectively. We found that six genes overlapped between the GWAS and DEGs gene sets in the tetrad stage (Figure 4b), including two downregulated genes (*LUSG00017676*, *LUSG00017705*) and four upregulated genes (*LUSG00017738*, *LUSG00017770, LUSG00028923, LUSG00017686*). Among them, the function of the *LUSG00017705* gene is cysteine synthase, which is the last key enzyme in the biological process of cysteine synthesis. The results of transcriptome analysis showed that the expression level of this gene in sterile flower buds was lower than that in fertile flower buds during the microspore stage (~3 times). The downregulated expression was also validated by qPCR and was significant (Student’s *t*-test, *p* < 0.01) (Appendix A). Fang et al. also obtained the same results in the study of pepper male sterility [30]. The high expression of the cysteine synthase gene can promote the synthesis of cysteine, and the increase in cysteine content promotes the synthesis of cysteine protease. Therefore, the expression of cysteine synthase is proportional to the expression of cysteine protease. We found a papain-like cysteine protease gene (*LUSG00017565*) in the GWAS analysis of sterile traits (Figure 4c), which is located in 24,467,740 bp to 24,473,461 bp on chromosome 10. It is 5722 bp in length, with 17 exons and 16 introns, encoding 737 amino acids. There are two mutations in this gene. One is located in the 1357 bp of the 13th CDS mutated from G to C, resulting in the mutation of methionine 519 of the protein to isoleucine; the other is distributed in the 6th exon. The mutation of A to C at 518 bp of the CDS resulted in the mutation of glutamic acid to proline at position 173 of the protein (Figure 4c). It is speculated that these two mutations lead to the early expression of the *LUSG00017565* gene in the tapetum cells during the meiotic tetrad formation of the pollen mother cells of the anthers, leading to the early apoptosis of the tapetum cells. As a result, microspores cannot be supplied with nutrients, resulting in pollen abortion. This indicated that the timely over-expression of the *LUSG00017565* gene is very necessary for the process of PCD and pollen development of the flax anther tapetum (Figure 4c).

Using the SNP loci significantly associated with male sterility to analyze the haplotype map, it was found that the heterozygosity of the sterile material was higher, and the fertile material was relatively homozygous. This is related to its biological background in which male sterile plants can only be pollinated by other varieties, and the recombined genes are separated into fertile and sterile plants through progeny. Fertile plants are homozygously stabilized by selfing; sterile plants continue to receive foreign genes, and the cycle repeats so that sterile material consistently remains highly heterozygous. It also illustrates the reliability of the male sterility-related SNP loci and candidate genes we detected in flax (Figure 5).

## 3. Discussion

By utilizing a combination of PacBio long reads, a genetic map, and Hi-C paired-end sequencing data, we successfully assembled a high-quality genome sequence of the flax Neiya No.9 cv. The assembled genome covered approximately 94.71% of the flax genome, totaling 473.55 Mb, and was anchored onto 15 chromosomes. The assembly demonstrated impressive statistics, with N50 lengths of 0.91 Mb for contigs and 31.72 Mb for scaffolds. Furthermore, we achieved a comprehensive annotation of 95.9% of the conserved BUSCO core gene set within the flax genome. Compared to previous studies [16,17], our assembly represents a substantial improvement in terms of assembly quality, particularly in the challenging task of assembling repetitive regions. This improvement is crucial for accurately characterizing the complex genome of flax, which has recently undergone whole-genome duplication and harbors a significant number of repeat elements. Compared to assembly based on short reads [16], our assembly using PacBio CLR sequencing data showed significant improvements in continuity and the integrity of repetitive sequences. However, when compared to assembly based on PacBio HiFi sequencing [19], it still lacks continuity. For further enhancement of sesame genome assembly quality, a combination of PacBio HiFi and ONT Ultra-long sequencing technologies should be considered [31]. Notably, our WGD analysis aligns with the results of Wang’s flax genome analysis [16], confirming the impact of repetitive sequences, especially long terminal repeats (LTRs), in the expansion of the flax genome. In addition to the genome assembly, we successfully annotated 32,786 protein-coding genes in the flax genome. Through gene family analysis, we identified specific orthogroups that are rapidly evolving, particularly those associated with fatty acid metabolism. These findings provide valuable insights into the genetic basis of flax and its potential applications in improving flax and other oilseed crops.

Male sterility has become the main direction and goal of crop utilization of heterosis and plays an important role in crop breeding and production [32]. In wheat, 42 varieties were cultivated using the male sterile *RMs2* hybrid line, with a cultivated area of 12.3 million hm^2^ and an increase of 5.6 million t in wheat yield [33]. In rice, the wild abortive cytoplasmic male sterile line is widely used in China and has achieved great success, with an increased yield of 20% to 30% [34]. The F1 generation of interspecific hybridization of flax has a heterosis rate of 25% to 40%, indicating that the use of heterosis can improve the economic benefits of flax [35]. However, flax is a highly self-pollinating crop, which requires manual emasculation in conventional breeding, which can lead to flower damage and yield decline and is time consuming and labor intensive. Therefore, obtaining flax male sterile material and mining related candidate genes is the key to the high-quality utilization of flax hybrids. The formation mechanism of male sterility in plants is relatively complex. The differentiation of sporogenous cells, the development of microspores and meiosis, and the abnormal differentiation of pollen or anthers can lead to male sterility [32]. Our morphological study on the male sterile material of flax found that the male sterile flowers of flax were divided into two types, namely open and non-open, and cytological studies showed that the pollen mother cells of sterile plants disintegrated prematurely from the first division of meiosis to the second mitosis, leading to pollen abortion.

The tapetum, the innermost cell of the anther sac, undergoes PCD during anther development, and the premature degradation of tapetum cells can lead to pollen abortion. We obtained four candidate genes by GWAS and RNA-seq analysis. Among them, the disfunction of *LUSG00017705* (cysteine synthase) has been proven to cause male sterility in many plants. However, we did not find any missense variant that occurred on this gene. Taken together with the result of transcriptome analysis, we hypothesize that the cause of male sterility in flax may not be the loss of the function of this gene. We suggest that cysteine synthase regulates the expression of cysteine protease genes by synthesizing cysteine, thereby regulating the timely disassembly of tapetum cells. Cysteine proteases play an important role in the PCD process of the tapetum [36,37]. Papain-like cysteine proteases have been found to be involved in the PCD process of the tapetum in a variety of plants, for example, in *NtCP56* in tobacco [36]; *Arabidopsis* cysteine protease 51 (*CP51*) [38], *NtCP56* [36], *CEP1* [39], and *βVPE* [40]; *Brassica napus BnaC.CP20.1* [41]; *OsCP1* [42] in rice, the abnormal expression of these protease genes affects the process of tapetum PCD, resulting in different degrees of pollen abortion. We found that the candidate gene *LUSG00017565*, which is significantly associated with male sterility in flax, belongs to the papain-like cysteine protease and is located at 24,467,740 bp to 24,473,461 bp on chromosome 10. This gene has two mutation sites, one at position 519, leading to the mutation of methionine to isoleucine, and the second one at position 173, which changes glutamine into proline. As a result, the expression time of the *LUSG00017565* gene in tapetum cells is out of balance, which leads to early PCD in tapetum cells, resulting in pollen abortion.

## 4. Materials and Methods

### 4.1. Plant Materials for Genome Assembly

The genome assembly of Neiya No.9 cv. (*Linum usitatissimum* L.), an inbred line of oil flax, was performed using PacBio long-read sequencing and Hi-C sequencing techniques. Young leaves of the plant were collected and rapidly frozen in liquid nitrogen upon collection. The frozen leaf samples were then stored at −80 °C until further processing. For RNA sequencing, various fresh plant tissues, including roots, stems, leaves, flowers, and fruits, were collected. To ensure sample purity, the collected plant tissues underwent a thorough washing process using ultrapure water, which was repeated three times. After washing, the plant tissues were immediately frozen in liquid nitrogen and stored at −80 °C for subsequent RNA extraction and sequencing.

The GWAS panel used in this study comprised 56 flax accessions, including 20 male sterility samples and 36 male fertility samples. The male fertile accessions were representative flax varieties grown in China and collected by our team. The male sterile accessions were derived through hybridization of the earliest sterile plants with other male fertile resources. They are inbred lines with relatively close genetic relationships. In order to validate the results obtained from genome-wide association studies (GWAS), we conducted a sample collection process targeting flower buds of both sterile and fertile plants. The collection was carried out at two specific developmental stages: the microspore mother cell stage and the tetrad stage. For each stage, we collected three independent biological replicates. Transcriptome sequencing was performed on these samples.

### 4.2. Library Construction and Sequencing

For the construction of the SMRTbell library, genomic DNA was extracted from the sample using the FastPure Plant DNA Isolation Mini Kit (Vazyme, Nanjing, China) following the manufacturer’s instructions. The DNA was then fragmented to an average size of 20 kb using a g-TUBE device from Covaris (Woburn, MA, USA). The fragmented DNA underwent end-repair, A-tailing, and ligation with hairpin adapters. Subsequently, the ligated fragments were purified and size-selected using BluePippin from Sage Science (Beverly, MA, USA) to retain fragments ranging from 10 kb to 50 kb. The resulting library was subjected to annealing, binding, and polymerase steps using the PacBio DNA/Polymerase Binding Kit P6 from Pacific Biosciences (Menlo Park, CA, USA). The bound complexes were purified, and annealing with sequencing primers was performed. Finally, the SMRTbell library was sequenced using the PacBio Sequel System, also from Pacific Biosciences (Menlo Park, CA, USA).

For the construction of short-read paired-end sequencing libraries, genomic DNA was isolated from the sample using standard methods. The DNA was fragmented using a Covaris M220 focused ultrasonicator (Covaris, Woburn, MA, USA). Subsequently, the fragmented DNA underwent end-repair, A-tailing, and adapter ligation using the KAPA HyperPrep Kit from KAPA Biosystems (Wilmington, MA, USA). Size selection of the fragments was performed using AMPure XP beads from Beckman Coulter (Brea, CA, USA). The resulting library was amplified and indexed using the KAPA HiFi HotStart ReadyMix, also from KAPA Biosystems (Wilmington, MA, USA). Quantification of the library was carried out using Qubit from Thermo Fisher Scientific (Waltham, MA, USA), and quality assessment was performed using an Agilent 2100 Bioanalyzer from Agilent Technologies (Santa Clara, CA, USA). Finally, the library was sequenced on the Illumina NovaSeq 6000 platform from Illumina (San Diego, CA, USA).

### 4.3. Genome Size Estimation

To model the k-mer count distribution and estimate the genome characteristics of Neiya NO.9 cv., we employed GenomeScope v1.0 [43]. The histogram of k-mer frequencies with a k-value of 111 was computed using jellyfish v1.1.12 [44].

To estimate the genome size, we utilized a BD Accuri C6 flow cytometer from BD Biosciences (San Jose, CA, USA). The excitation and emission wavelengths were set to 488 nm and 695 nm, respectively. Fresh leaves from young plants were collected, and nuclei were isolated following a standard protocol. The stained nuclei were analyzed using the CFlow software (BD Biosciences, San Jose, CA, USA). Data with a coefficient of variation below 4% were considered for genome size estimation. The formula used to calculate the genome size was as follows: genome size = (sample mean/standard reference mean) × standard reference genome size. In this study, we adopted the genome size of Arabidopsis thaliana as the standard reference, which is 125 Mb.

### 4.4. Genome Assembly

Long-read assembling was performed by using MaSuRCA v3.3.1 [45]. Then, the Hi-C sequencing data were mapped to the contig using BWA-mem v0.7.12 [22], and the Hi-C contact map was generated using the 3D-DNA v180922 [23] pipeline. A previously reported genetic map constructed using an F2 population was also used [21]. ALLMAPS v0.9.14 [46] was used to anchor contigs to chromosomes by integrating the evidence from Hi-C and the genetic map.

### 4.5. Genome Annotation

For de novo identification of repeat families, we utilized RepeatModeler v2.0.1 [47], which employs RepeatScout v1.0.6 [48] and RECON v1.08 [49]. The TEs were identified using LTRharvest [50] from the GenomeTools v1.6.1 [51] and LTR_retriever v2.8-0 [52] tools. A redundancy removal process was applied to discover high-quality LTR families. The consensus sequences of TE families were classified using both the Dfam v3.1 [53] and Repbase v20181026 [54] databases. RepeatMasker v4.0.9 [55] was used with the custom repeat library to identify repeat sequences.

For gene prediction using RNA-Seq data, we mapped the sequencing data to the flax genome using HISAT2 v2.1.0 [56] and assembled the mapped reads using Stringtie v2.1.1 [57]. The pipeline for ab initio gene annotations involved de novo gene predictions of the repeat-masked genome using AUGUSTUS v3.4.0 [58] and SNAP v2013-02-16 [59]. Additionally, evidence-based gene annotations were performed using MAKER2 v2.31.10 [60].

To functionally annotate the genes, we conducted BLASTP v2.9.0 [61] analysis against the SwissProt [62] and NR databases with an E-value threshold of 1 × 10^−5^. InterProScan v5.41-78.0 [63] and HMMER analyses were performed against the InterPro and Pfam databases, respectively. tRNA genes were identified using tRNAscan-SE v1.3.1 [64], while rRNA fragments were predicted through alignment to rRNA sequences using BLASTn with an E-value of 1 × 10^−10^. miRNAs and snRNAs were predicted using INFERNAL v1.1.2 [65] and the Rfam [66] database. Whole-genome-duplication (WGD) analysis was conducted using MCScanX v2020.8.18 [67] software with default parameters.

### 4.6. Gene Family

Gene family analysis was conducted using OrthoFinder v2.4.0 [68]. The analysis involved identifying single-copy gene families, which were then utilized to construct a species tree. First, each orthogroup was aligned using MUSCLE v3.8.1551 [69]. The resulting alignments were concatenated to construct a maximum-likelihood phylogenetic tree using IQ-TREE v2.0.3 [70] with default parameters. The branch lengths of the species tree were calibrated using calibration points obtained from TimeTree [71] and integrated into the tree using r8s v1.8.1 [72]. To model the expansion and contraction of orthologous gene families, CAFE v4.2.1 [73] was employed. This software enabled the estimation of the gain and loss of genes within these gene families.

### 4.7. Transcriptome Analysis

To extract total RNA, the Trizol reagent from Invitrogen (San Diego, CA, USA) was used following the manufacturer’s instructions. During the extraction process, DNase was added to remove DNA. The quality of the RNA samples was assessed on 1% agarose gels, and the RNA integrity values (RIN) were determined using the Agilent 2100 Bioanalyzer from Agilent Technologies (Santa Clara, CA, USA). For cDNA library construction, 2 μg of RNA per sample was utilized. The NEBNext Ultra™ RNA Library Prep Kit for Illumina from NEB (Ipswich, MA, USA) was employed according to the manufacturer’s instructions to generate sequencing libraries. Library quality was assessed using the Agilent Bioanalyzer 2100 system. The prepared libraries were then sequenced on an Illumina NovaSeq 6000 platform using a paired-end read protocol with 150 bp of data collected per run.

The raw reads underwent trimming and filtering using Cutadapt v2.8 [74]. Subsequently, they were mapped to the Neiya No. 9 genome assembly using HISAT2 v2.2.1 [56]. Reference-guided transcript assembly was performed using Stringtie v2.1.4 [57], and read counts for each gene were measured using Ballgown v1.99.6 [75]. Differential gene expression analysis was conducted using DESeq2 v1.31.2 [76]. Genes exhibiting an absolute log2 (fold-change) > 1 and a q-value (false discovery rate (FDR)) < 0.01 were identified as differentially expressed genes (DEGs). The R package ClusterProfiler [77] was employed for GO enrichment analysis.

### 4.8. Sequence Alignment and Variant Detection

For the GWAS, the paired-end sequencing reads were aligned to the genome assembly using BWA mem [22] with default parameters. Picard Tools, available at http://broadinstitute.github.io/picard/ (accessed on 13 Feburay 2021), was utilized to coordinate, sort, and remove PCR duplicates, resulting in a BAM file. An index for the BAM file was also generated. Genome variants were called using the Genome Analysis Toolkit (GATK) [78]. Low-quality variants were filtered based on specific criteria. For SNPs, variants with QD < 2.0, MQ < 40.0, FS > 60.0, MQRankSum < −12.5, or ReadPosRankSum < −8.0 were filtered out. InDels were filtered based on QD < 2.0, FS > 200.0, or ReadPosRankSum < −20.0. To aid in gene-based SNP annotation, SnpEff [79] was employed.

### 4.9. Observation of Paraffin Sections from Normal and Sterile Flower Buds

Fixation, Dehydration, and Staining. Flower buds from both fertile and sterile flax plants at different developmental stages were selected. Quickly place the buds in a small bottle with FAA fixative solution in a 20–30 times ratio. Cover the bottle, label it, and store it at 4 °C. Dehydrate the buds by sequentially incubating them in 30% ethanol for 30 min, 50% ethanol for 30 min, and then stain them with hematoxylin and eosin staining solution for 2 days. Wash the material with distilled water until the color disappears, then with tap water until it turns dark blue. Subsequently, incubate the buds in 70% ethanol for 30 min, 80% ethanol for 30 min, 90% ethanol for 30 min, and pure ethanol for 30 min overnight. Use 20 times the volume of the material for each ethanol step.

Clearing, Embedding. Clear the dehydrated flower buds using xylene for 2 h, following the steps of: 3:1 ethanol:xylene for 2 h, 1:1 ethanol:xylene for 2 h, 1:3 ethanol:xylene for 2 h, xylene for 1 h, and xylene for 30 min. Then, embed the buds in molten paraffin wax after being floated on xylene with paraffin wax chips at 45 °C for 3 h until the saturation of paraffin wax. The temperature should be maintained at 60 °C whilst being changed into pure paraffin wax twice for 30 min each time overnight.

Sectioning, Mounting, and Drying. Before sectioning, cut the paraffin blocks into square or rectangle pieces, leaving a 3 mm width of paraffin around the tissue. Put the paraffin block on the slicing machine at an angle of about 15° and use the slicer to obtain 8 μm sections. Place the sections on a clean microscope slide containing distilled water and let the paraffin float on the surface of the water. Absorb the excessive water with filter paper, transfer the section onto a drying bench at 45–48 °C, and dry the sections until they are flattened. Deparaffinize by immersing the slides in xylene for 15 min and mount the sections using neutral gum.

Dewaxing and Sealing. Dewax the slides by immersing them in xylene in a staining dish facing the flower bud section upwards for 15 min. After dissolving the paraffin wax completely, seal the slides with neutral gum.

Observation and Photography. Place the sealed slides flat and dry, then observe the flower bud sections under a microscope and take photographs.

### 4.10. Genome-Wide Association Analysis

For the GWAS on male sterile plants, linear mixed models were utilized, and the analysis was conducted using the GEMMA v0.98.1 [80] software. GEMMA is a software implementing the Genome-wide Efficient Mixed Model Association algorithm for a standard linear mixed model for genome-wide association studies. It fits a standard linear mixed model (LMM) to account for population stratification and sample structure for single-marker association tests. In the analysis, the kinship was included as a random effect to account for genetic relatedness among individuals, and the first three principal components from PCA analysis were included as fixed effects to account for population structure. We filtered out SNPs with a minor allele frequency (MAF) of less than 0.05 and a missing rate greater than 10%. To determine significant associations, a threshold for *p*-values was set using the Bonferroni correction. The significance threshold was calculated as 0.05 divided by the number of SNPs tested (n). Therefore, if the number of SNPs tested was denoted as n, the threshold for significance was defined as 0.05/n. For genome-wide significance, the calculated threshold was *p* = 2.2 × 10^−8^, meaning that associations with a *p*-value below this threshold were considered statistically significant at the genome-wide level.

## 5. Conclusions

In summary, our work has successfully generated high-quality assembly of the oil flax genome and established reliable gene annotations. This achievement lays a strong foundation for various areas of research, such as functional genomics, comparative genomics, and molecular breeding of flax. Additionally, through the use of GWAS and RNA-Seq analysis, we have identified potential candidate genes that could be involved in the male sterility trait in flax. This finding is significant as it opens up possibilities for further investigations and understanding of the underlying mechanisms of male sterility. Moreover, the generated resources and knowledge will greatly facilitate hybrid breeding research in flax, leading to improved cultivars and enhanced agricultural practices in the future.

## Figures and Tables

**Figure 1 plants-12-02773-f001:**
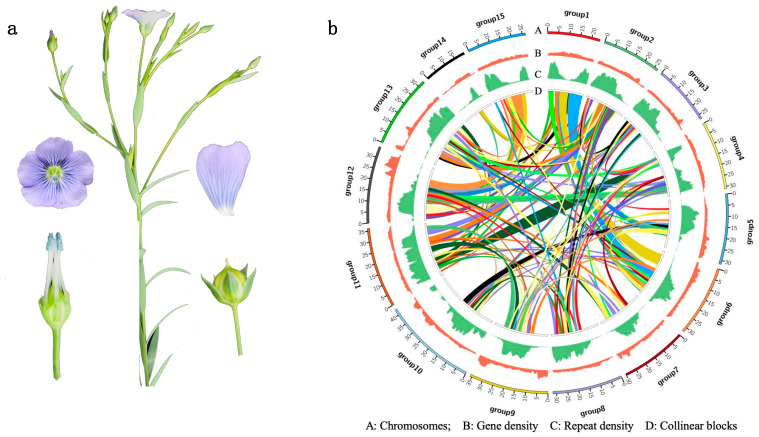
Morphological characteristics of the Neiya No. 9 cv. (**a**). Genomic features and collinear blocks across the Neiya No. 9 cv. genome assembly (**b**).

**Figure 2 plants-12-02773-f002:**
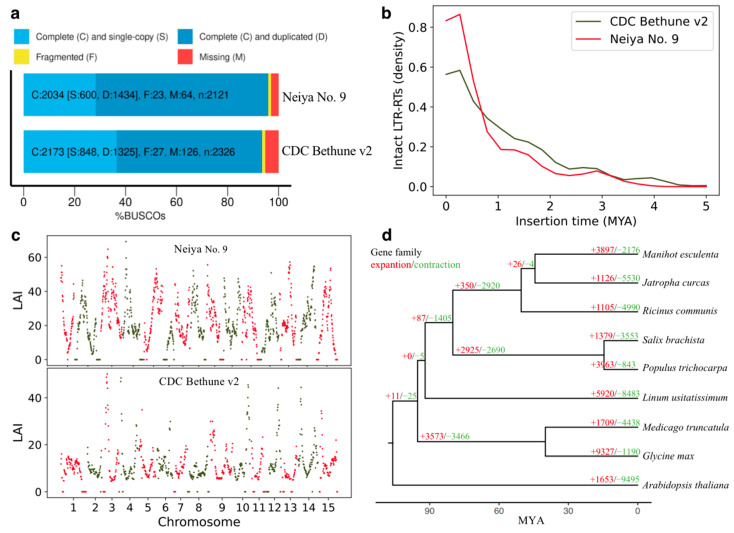
Genome assembly quality assessment. (**a**) BUSCO assessment for genome assemblies of Neiya No. 9 cv. and CDC Bethune v2. (**b**) Estimated insertion times for intact LTR-RTs (less than or equal to 5 million years ago) in the genome assemblies of Neiya No. 9 and CDC Bethune v2. (**c**) Evaluation of Neiya No. 9 and CDC Bethune v2 genome assemblies using the LTR Assembly Index (LAI). The red and dark green dots represent the LAI values of different chromosomes in 3Mb sliding windows. (**d**) Gene family characteristics comparison between *Linum usitatissimum* and eight other dicot plants.

**Figure 3 plants-12-02773-f003:**
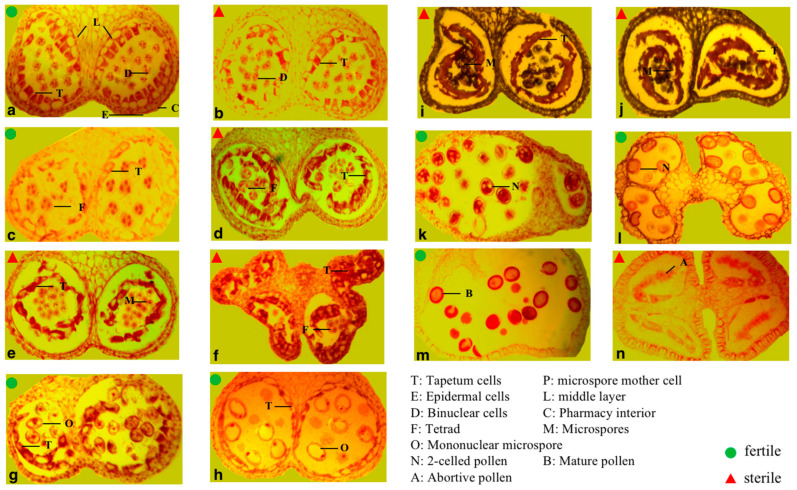
Comparison of pollen development between fertile and infertile flax. (**a**) The microspore mother cell of fertile plant. (**b**) Asynchronous development of the adjacent two anther chambers at microspore mother cell stage of sterile plant. (**c**) The cross-section of anther at the dyad stage of fertile plant. (**d**) The dyad stage of sterile plan. (**e**) The cross-section of anther showing anther wall and tetrad of fertile plant. (**f**) The tetrad stage in the cross-section of anther of sterile plant. (**g**) The cross-section of anther showing the formation of microspore of fertile plant. The nucleus was located at the center. (**h**) The cross-section of anther showing that the nucleus was close to the wall and the tapetum degenerated in fertile plant. (**i**) The cross-section of anther of sterile plant. The development of microspore stagnated. The tapetum cell wall dissolved and the nucleus degenerated. The shrunk protoplasm surrounded microspores. (**j**) In sterile plant, degenerated microspores aggregated, surrounded and squeezed to the side by protoplasm, in which a great deal of vesicle occurred. (**k**) The two-celled pollen stage of fertile plant. (**l**) The cross-section of the near-mature anther showing anther wall and pollen grains of fertile plant. (**m**) The longitudinal section of mature anther showing epidermis, fibrous layer, and mature pollen grains of fertile plant. (**n**) The aborted anthers of sterile plant. The deep colored part in the anther chamber indicated the residue of tapetum and aborted pollen.

**Figure 4 plants-12-02773-f004:**
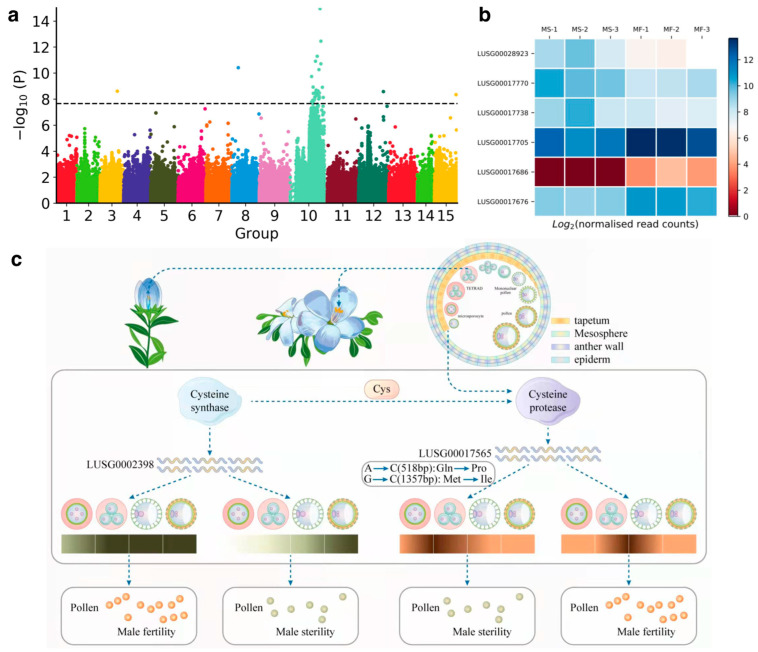
Genome-wide associations study of male sterility (**a**). Transcriptome verification of candidate genes (**b**). Schematic diagram of the mechanism of male sterility (**c**).

**Figure 5 plants-12-02773-f005:**
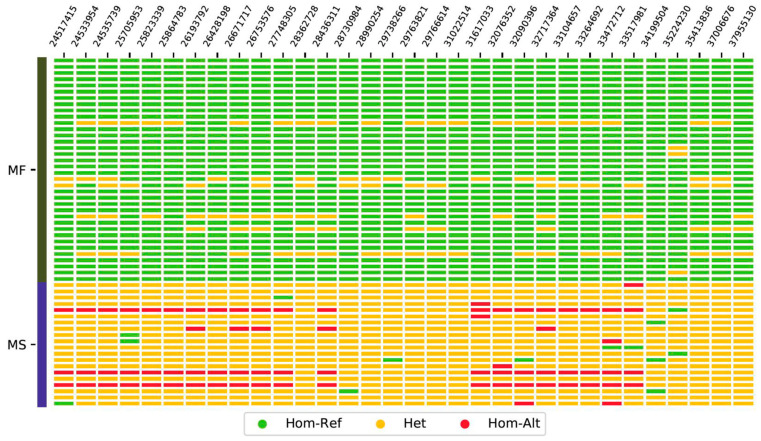
Haplotypes were constructed using significant SNPs in the candidate region. The numbers at the top of the graph indicate the positions of the SNPs on chromosome 10.

## Data Availability

The raw sequence data generated in this study have been deposited in the Genome Sequence Archive [81] at the National Genomics Data Center (NGDC) [82] in China. The accession number for the data is GSA: PRJCA009864, and it can be accessed at https://ngdc.cncb.ac.cn/gsa (accessed on 2 May 2022). The RNA-Seq data associated with this study can be downloaded from the GenBank under the project ID PRJNA725803. The assembly and annotation files for the flax genome are available at Zenodo and can be accessed at https://doi.org/10.5281/zenodo.7811972 (accessed on 8 Aprl 2023). The genomes used for gene family analysis were obtained from Phytozome version 12, and the data can be accessed at https://phytozome-next.jgi.doe.gov/ (accessed on 7 August 2020).

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
