# Peer review of "High-Quality Genome Assembly and Genome-Wide Association Study of Male Sterility Provide Resources for Flax Improvement"

_plants, 2023, doi:10.3390/plants12152773_

Round 1

Reviewer 1 Report

In line 504, the authors mentioned that GEMMA was used for GWAS analysis. There are numerous GWAS statistical methods. Could you explain why this method was used?

Could you think that the inclusion of other GWAS methods may be more convincing? If a SNP is detected by multiple methods, it is more likely to be associated with the target trait.

Do you think if it is better to give some brief introduction of the statistical model of GEMMA?

Some details should be added such as the Minor allele frequency (MAF) in the GWAS analysis. If there are missing genotype for one loci, please explain the treatment in this situation.

Minor editing of English language required.

Reviewer 2 Report

The subject is interesting and a valuable resource for the community and should be published after revision.

Specific comments:

Introduction:

- Line 46ff: as a major rationale of the manuscript is the molecular mechanism of male sterility, authors should explain the heterosis effect as the rationale of investigating male sterility clearly at the beginning.    

- Line 54 ff: “….were also not successful in its utilization for hybrid production.”

Results:

- Line 101 ff: it is reasonable to compare sequence quality parameters with the CDC Bethume v2 assembly from 2018, as this is the best other oil seed flax variety sequence. The manuscript however would gain, if a comparison with the sequence of the recent fibre flax variety sequence of Sa et al. 2021 would also be included for completeness and for a comparison of sequences that were generated by similar techniques.

- line 148 ff: can you please specify in more detail, what these “10150 groups of homologous gene families” are? Or are they groups of homologous genes?  And did you identify them?

- line 161 ff – Comparative genomics: Can authors please explain their rationale, why these eight species were taken for comparative analysis? Why did authors not compare oil with fibre flax sequence?

- line 178 ff /Fig 2d: Gene family expansion and contraction is shown with respect to L. usitatissimum. In A.thaliana the number of changed gene families is more than 11000 families according to the figure. How is this possible, if A.thaliana only has a total of less than 10000 gene families? OR do authors show gene number?

Is there anything to say about the nature/ function of the altered gene families and the comparative species?

- line 180: “… analysis identified x expanded and x … genes in flax”, in which comparison?

- Figure 2: font in all panels is much too small and not readable on a print out, please increase. Also, move text in panel C to somewhere, where letters do not cover data points in the figure. Panel d also here numbers are much too small

- Line 225: can authors say anything about the rationale, why they chose these particular 20 sterile and 36 fertile genotypes for GWAS analysis?

- line 234: what is the rationale of doing GO analysis of the group of genes that harbour significantly associated SNPs in GWAS across a region of 18.8 MB?

Would it not make more sense to do GO analysis in the set of genes that are differentially expressed in the performed RNAseq experiment? (line 246 ff)

-          Line 247ff: Can authors be more specific in describing the samples used and compared to each other?

-          Line 250ff: it is a satisfying analysis to determine the overlap between GWAS hits and differentially expressed genes, however the genes that are causal for male sterility may well be upstream of the genes that are differentially expressed.

-          Line 257: “The downregulated expression….”

-          Line 276: authors find that SNPs that are significantly associated with male sterility are more often in the heterozygous state in sterile material. Is this not true for all of the SNPs genome-wide?

-          Figure 5: would it not make more sense to use a barplot to image a comparison of the levels of het, homo-Ref, homo-alt between MS and MF in sign SNPs as well as genome-wide?

Discussion:

-          Also include comparison of Sa et al 2021 genome quality which is also made with the same sequencing technology. It is very valuable to have several high quality flax genomes, especially one oil seed and one fibre.

Methods:

-          Transcriptom analysis: if RNA extraction was done as stated in the protocol - with TRIZOL and without DNase treatment, data are likely to be contaminated with genomic DNA and may not be meaningful. This however should have been note with the Bioanalyzer?

-          Line 371: please specify which standard DNA isolation method was used.

-          Line 506: for GWAS analysis kinship was used as random factor in the model. Why did authors use PCA 1-3 as fixed effects?
